# Evaluation of Oxidative Stress and Metabolic Profile in a Preclinical Kidney Transplantation Model According to Different Preservation Modalities

**DOI:** 10.3390/ijms24021029

**Published:** 2023-01-05

**Authors:** Mrakic-Sposta Simona, Vezzoli Alessandra, Cova Emanuela, Ticcozzelli Elena, Montorsi Michela, Greco Fulvia, Sepe Vincenzo, Benzoni Ilaria, Meloni Federica, Arbustini Eloisa, Abelli Massimo, Gussoni Maristella

**Affiliations:** 1Institute of Clinical Physiology, National Research Council (IFC-CNR), 20159 Milano, Italy; 2Department of Molecular Medicine, IRCCS Foundation Policlinico San Matteo, 27100 Pavia, Italy; 3Department of Surgery, IRCCS Foundation Policlinico San Matteo, 27100 Pavia, Italy; 4Department of Human Sciences and Promotion of the Quality of Life, San Raffaele Roma Open University, 00166 Roma, Italy; 5Institute of Chemical Sciences and Technologies “G. Natta”, National Research Council (SCITEC-CNR), 20133 Milan, Italy; 6Section of Pneumology, Department of Internal Medicine, University of Pavia, 27100 Pavia, Italy; 7Centre for Inherited Cardiovascular Diseases, IRCCS Foundation Policlinico San Matteo, 27100 Pavia, Italy

**Keywords:** kidney transplant, hypothermic machine perfusion, organ preservation, ROS, metabolomic, EPR, ^1^H-NMR, oxidative damage

## Abstract

This study addresses a joint nuclear magnetic resonance (NMR) and electron paramagnetic resonance (EPR) spectroscopy approach to provide a platform for dynamic assessment of kidney viability and metabolism. On porcine kidney models, ROS production, oxidative damage kinetics, and metabolic changes occurring both during the period between organ retrieval and implantation and after kidney graft were examined. The ^1^H-NMR metabolic profile—valine, alanine, acetate, trimetylamine-N-oxide, glutathione, lactate, and the EPR oxidative stress—resulting from ischemia/reperfusion injury after preservation (8 h) by static cold storage (SCS) and ex vivo machine perfusion (HMP) methods were monitored. The functional recovery after transplantation (14 days) was evaluated by serum creatinine (SCr), oxidative stress (ROS), and damage (thiobarbituric-acid-reactive substances and protein carbonyl enzymatic) assessments. At 8 h of preservation storage, a significantly (*p* < 0.0001) higher ROS production was measured in the SCS vs. HMP group. Significantly higher concentration data (*p* < 0.05–0.0001) in HMP vs. SCS for all the monitored metabolites were found as well. The HMP group showed a better function recovery. The comparison of the areas under the SCr curves (AUC) returned a significantly smaller (−12.5 %) AUC in the HMP vs. SCS. EPR-ROS concentration (μmol·g^−1^) from bioptic kidney tissue samples were significantly lower in HMP vs. SCS. The same result was found for the NMR monitored metabolites: lactate: −59.76%, alanine: −43.17%; valine: −58.56%; and TMAO: −77.96%. No changes were observed in either group under light microscopy. In conclusion, a better and more rapid normalization of oxidative stress and functional recovery after transplantation were observed by HMP utilization.

## 1. Introduction

Kidney transplantation can be considered the best treatment for end-stage renal failure, with longer life expectancy and superior quality of life [1,2,3].

However, the lack of suitable organ donors is a major constraint to transplantation.

Thus, to decrease the disparity in supply and demand, to the Standard Criteria Donors (SCD), the Donation after Cardiac Death (DCD) [4] and the Expanded Criteria Donor (ECD) [5,6,7,8] groups have been added [9]. This implies that organs are most often in marginal conditions because the donors present co-morbidity factors (diabetes, obesity, cardiovascular disease, and/or hypertension). These factors weaken the organ, making it more susceptible to developing lesions correlated to the graft outcome. To overcome these problems, innovative means have been established to obtain kidneys suitable for transplantation and, at the same time, to be able to determine the kidney quality. From this new perspective, the effort is devoted to assessing the best method of organ preservation while at the same time, developing methodologic criteria leading to the determination of the reimplant suitability or discharging.

Successful transplantation depends on a sequence of events related to donor selection, graft preservation, surgical implantation, and recipient treatment during which the kidney is exposed to a series of non-physiologic insults. Particularly, at the time of organ retrieval from the donor, blood supply is interrupted, leading to ischemic insults to renal function. Thus, kidneys from all donor types are exposed to ischemia. However, the cause of ischemia differs according to the donor source: in the after-cardiac-arrest donor group, the ischemic period is divided into warm and cold ischemic times. In the first period, removal of the kidney from the circulatory system leads to the absence of oxygen and anaerobic metabolism [10,11].

Nonetheless, even under hypothermic conditions—the metabolic rate at levels of approximately 10%—the need for oxygen persists, and hypoxia remains the main source of lesions induced in the context of preservation. This is termed cold ischemic injury together with a strong catalyst for the generation of oxygen free radicals [12]. In synthesis, warm ischemia leads to hypoxic lesions, while cold ischemia leads to both hypoxic and hypothermic lesions that strongly correlate with immediate and long-term kidney function [13,14,15,16].

In all cases, ischemia is followed by reperfusion, which occurs when the graft is connected to the recipient vascular system, and this is well known to exacerbate the cellular injuries initiated by ischemia. Ischemia–reperfusion is a complex pathophysiological process involving hypoxia and/or reoxygenation, ionic-imbalance-induced edema and acidosis, oxidative stress, mitochondrial uncoupling, coagulation, and endothelium activation that is associated with a proinflammatory immune response [17,18].

Particularly, there is strong evidence that reactive oxygen species (ROS) are important mediators of ischemia reperfusion injury in organ transplantation [19,20]. Indeed, the observed oxidative burst triggers inflammation and tubular cell injury. ROS–molecule interactions, including (a) oxidation of amino acids, resulting in the loss of important functional properties; (b) lipid peroxidation of cell membranes [20], resulting in decreased membrane viability; and (c) cleavage and crosslinking of renal DNA, resulting in harmful mutations, promote renal injury through damage to molecular kidney components. Lipid oxidation gives rise to malondialdehyde (MDA), which is a good indicator of nephrotoxicity and tissue damage and is a reliable diagnostic biomarker of initial graft injury and dysfunction earlier than serum creatinine. Moreover, levels of MDA in plasma within the first week after reperfusion of the graft predict long-term graft outcome [21].

Static cold storage (SCS) is undoubtedly the simplest and most widely adopted method of hypothermic preservation [22,23]. 

However, with the increasing use of marginal kidneys, there has been a renewed interest into the use of hypothermic machine perfusion (HMP) [24,25,26]. The evidence suggests that HMP may be more beneficial in reducing delayed graft function (DGF) rates in marginal kidneys [27]. A recent study by Nath et al. [28], comparing porcine kidneys’ metabolic profiles, suggested that beneficial effects exerted by HMP might be related to the removal of ROS.

As is well known, electron paramagnetic resonance (EPR) is the only technique capable of providing direct quantitative detection of the ‘instantaneous’ presence of ROS [29,30] in a variety of biological samples [31,32,33,34,35], resulting in its being the “gold standard”.

At the same time, nuclear magnetic resonance is the technique of choice to obtain metabolic profiles in the field of medicine, biology, and food science [36,37,38,39,40].

By a joint EPR and ^1^HNMR analysis, the present study aims to examine the ROS production, the oxidative damage kinetics, and the metabolic changes occurring both during the period between organ retrieval and implantation and after kidney graft, according to the mode of preservation of the transplanted organ. 

All experiments were carried out on a porcine model that, for its intrinsic characteristics, is well adapted to the modeling of kidney transplantation [15,41,42,43]. ROS production, oxidative damage kinetics, and metabolic changes occurring both during the period between organ retrieval and implantation and after kidney graft were examined. A better and more rapid normalization of oxidative stress and functional recovery after transplantation were observed by HMP utilization.

## 2. Results

### 2.1. NMR and EPR Procedure

A joint use of the NMR and EPR techniques constitutes one of the highlights of the present study. A sketch of the procedures and typical EPR and ^1^H NMR spectra acquired from perfusate and tissues showing the main assigned resonances are displayed in Figure 1.

### 2.2. Analysis during the Preservation Procedure

The intra-renal resistance (IRR) resulted in the physiological range throughout the duration of time of the HMP preservation [44]. The recorded data are reported in Figure 2A. Starting from 4 mmHg/mL·min, the IRR values gradually but significantly (range *p* = 0.05–0.001) decreased, following a sigmoidal trend (R^2^ = 0.998), reaching a value of 0.75 ± 0.14 mmHg/mL·min at 4 h, then slowly decreasing up to the final level of 0.51 ± 0.12 mmHg/mL·min at the end of the perfusion.

The changes in the ROS production rate recorded in perfusion solution during HMP perfusion and in liquid storage solution during SCS are reported in Figure 2B. As shown in the figure, during the HMP perfusion, the ROS production rate data, after 15 min from the beginning, began to linearly increase at a rate of 0.054 ± 0.007 μmol·min^−1^ (R^2^ = 0.95), reaching the peak value of 0.099 ± 0.015 μmol·min^−1^ at 1 h, then slowly returning toward the initial level of 0.044 ± 0.003, following a single exponential trend (K = 0.1072, R^2^ = 0.99). All data resulted in being significantly (range *p* < 0.05–0.0001) higher than the initial level. Significant (*p* < 0.0001) increases of the ROS production rate with respect to the initial level were likewise measured at 4 (0.067 ± 0.012 μmol·min^−1^) and 8 h (0.087 ± 0.009 μmol·min^−1^) of SCS storage. As is shown in the figure, at 15 min and 4 h, the SCS data did not significantly differ from those measured in HMP. However, at 8 h of storage, a significantly (*p* < 0.0001) higher ROS production was measured in the SCS compared with the HMP group (0.056 ± 0.003 vs. 0.087 ± 0.009 μmol·min^−1^).

Antioxidant capacity data recorded during HMP perfusion and in SCS storage are reported in Figure 2B. Showing a specular trend with respect to the ROS production data reported in 1B, HMP values started to significantly and linearly decrease at a rate of 28.8 nW/h (R^2^ = 0.95) after 30 min of perfusion, that is, a little later than ROS (initial level 91.0 ± 2.0 nW), reaching a minimum at 1 h (76.0 ± 4.0 nW), then slowly returning toward the initial levels, in turn, following a single exponential trend (K = 0.4023, R^2^ = 0.98). All data recorded between 30 min and 4 h resulted in being significantly (range *p* < 0.05–0.001) lower than the initial value. A significant (*p* < 0.01) decrease of the TAC was recorded at 4 h in SCS (88.0 ± 3.0 nW) as well. In a similar fashion, as seen in Figure 2C, at 8 h of storage, the TAC measured from the SCS group resulted in being significantly (*p* < 0.001) lower than that of the HMP group (89.0 ± 2.0 vs. 97.0 ± 2.0 nW).

From a comparison of the ROS production rate increase with respect to the contemporary TAC decrease, during the first hour of HMP perfusion, the ROS level was found to have increased by approximately 60%, while the TAC decreased by approximately 23%. At the same time, at the end of the storage, the difference in the ROS levels measured from the SCS vs. HMP groups was approximately 35%, while the TAC level resulted in being approximately 8.2% lower.

#### ^1^H-NMR Analysis from Preservation Solutions

The absolute concentration (mM) data of the most important metabolites calculated by the ^1^H-NMR spectra recorded from the preservation solution during 8 h of storage at 4 °C for both the HMP and SCS groups are reported in Figure 3: lactate (A), trimethylamine *N*-oxide (TMAO, B), valine (C), alanine (D), and acetate (E). 

All these metabolites present de novo (therefore, likely produced by the kidney) in the solution showed an overall increase over time (T0 vs. T8): lactate: 0.0168 ± 0.0023 vs. 0.8440 ± 0.0300; TMAO: 0.0018 ± 0.0004 vs. 0.0500 ± 0.0081; valine: 0.0011 ± 0.0001 vs. 0.0176 ± 0.0002; alanine: 0.0014 ± 0.0001 vs. 0.0470 ± 0.0037; and acetate: 0.0185 ± 0.0028 vs. 0.0211 ± 0.0135. Moreover, as is shown in the figure, a significantly higher concentration data (*p* < 0.05–0.0001) was calculated for the HMP with respect to the SCS group during the entire experimental time and for all the monitored metabolites. In a similar fashion, the amount of all metabolites in the HMP group suddenly increased during essentially the first hour of storage, following a single exponential trend, even though at a different rate (see Figure 3). The rate of concentration change was greater in HMP kidneys than those of the SCS group, where a significant increase (*p* < 0.05–0.01) of the metabolite concentration (mM) was observed during the 8 h of storage (T0 vs. T8): lactate (0.017 ± 0.002 vs. 0.220 ± 0.051), TMAO (0.0019 ± 0.0019 vs. 0.0120 ± 0.0065), valine (0.0018 ± 0.0005 vs. 0.066 ± 0.0012), alanine (0.0016 ± 0.0001 vs. 0.0107 ± 0.0059), and acetate (0.0189 ± 0.0094 vs. 0.0384 ± 0.0194).

Finally, glutathione compound was one of the most important constituents of both HMP and SCS preservation solutions. The total glutathione (GSH) concentration levels calculated during the HMP and SCS storage are reported in Figure 3F. As is shown in the figure, a significant (*p* < 0.05–0.0001) single exponential GSH consumption was found during the HMP (T0 1.95 ± 0.14 vs. T8 0.47 ± 0.07 mM), while throughout the 8 h of SCS storage (T0 1.88 ± 0.28 vs. T8 1.67 ± 0.37 mM), the level remained almost unchanged.

Finally, positive correlations were found between ROS production by EPR and NMR metabolites: lactate (r = 0.97, *p* = 0.0008), TMAO (r = 0.94, *p* = 0.004), valine (r = 0.95, *p* = 0.003), alanine (r = 0.96, *p* = 0.001), and acetate (r = 0.88, *p* = 0.02), while inverse correlation was found with oxy-GSH (r = −0.94, *p* = 0.004).

### 2.3. Functional Analysis during the Survival Period

As an inclusion criterion, in the present study, it was established that the animal had to survive without any complications until the 14th day after kidney transplantation. Therefore, all animals that prematurely died or showed any type of complication during this period were excluded from the study. Indeed, all animals belonging to the HMP group reached the end of the follow up, while only five animals of the SCS group survived. One animal death was attributable to a renal failure possibly related to primary non-function (PNF), and the other animal died after a massive haemorrhage due to the traumatic removal of the jugular central venous line produced by the animal itself.

#### 2.3.1. Early Functional Recovery

The functional recovery after transplantation was evaluated by serum creatinine concentration assessment (SCr). Figure 4A shows the SCr values obtained from the HMP (dashed line) and SCS (continuous line) groups during the follow-up period. No SCr differences were found between HMP and SCS at Day 1. In the following days, until the sacrifice, compared with the animals of the SCS group, those belonging to the HMP group showed a better function recovery, even though the levels of the SCr peak (approximately 12.7 mg/dL) were similar and were reached at approximately the same time (5th day). The comparison of the areas under the curves (AUC) returned a significantly smaller (−12.5%) AUC in the HMP compared with the SCS group. 

#### 2.3.2. Oxidative Stress Evaluation

The plasmatic concentration levels of ROS, TBARS, and PC, measured during the post transplantation, are, respectively, shown in Figure 4B–D for the HMP (empty symbols) and SCS (full symbols) groups. 

As can be observed in Figure 4B, the ROS production rate (μmol·min^−1^) started from a higher level and remained significantly higher (range *p* < 0.05–0.0001) during the entire period in the SCS in comparison with the HMP group. Starting from a concentration of 0.14 ± 0.04 and 0.12 ± 0.01 μmol·min^−1^ for the SCS and HMP, respectively, the differences became greater in the following days, reaching a peak at the 4th day, then the values slowly returned, measuring below the initial levels at the 14th day: SCS 0.13 ± 0.04 and HMP 0.11 ± 0.01 μmol·min^−1^. 

Moreover, with respect to the 1st day, significant increases were found within each group as follows: SCS: at 3rd day 0.23 ± 0.09, 4th day 0.24 ± 0.06, and 5th day 0.22 ± 0.06; HMP: at 3rd day 0.21 ± 0.03 and 4th day 0.20 ± 0.06.

As is shown in Figure 4C, an even greater difference of the values of TBARS concentration (μM), a biomarker of lipids peroxidation, recorded in the two groups was found. With respect to Day 1, SCS: 8.45 ± 0.78 and HMP: 5.75 ± 0.63 were reported during the monitored period; the data calculated for the HMP group resulted in being significantly (range *p* < 0.01–0.0001) lower compared with the other group.

At the same time, with respect to the 1st day, significant increases (range *p* < 0.05–0.0001) were found within each group, reaching a peak at the 4th day, as follows: SCS: at 2nd day 11.26 ± 0.53, 3rd day 13.24 ± 0.53, 4th day 14.30 ± 1.71, 5th day 14.76 ± 1.99, 6th day 13.29 ± 1.55, and 7th day 11.96 ± 1.45, thereafter slowly returning to the initial level at the 14th day 9.37 ± 0.64. HMP: 3rd day 8.45 ± 0.53, 4th day 11.15 ± 0.84, 5th day 9.38 ± 0.58, 6th day 8.30 ± 1.55, and 7th day 8.52 ± 1.45, thereafter slowly returning toward the initial level at 14th day 7.83 ± 0.64 as well.

Regarding PC (nmol·mg^−1^ protein), a biomarker of proteins oxidation, an almost similar trend can be observed for the two groups within the first 4 days (Figure 4D). Then, the difference became greater, reaching a peak at the 5th day. Thereafter, the values slowly returned towards the initial level at the 14th day: SCS: 1.01 ± 0.31 and HMP: 0.96 ± 0.07, even though the data recorded for the HMP treated kidneys resulted in being lower than for the SCS group. Significant differences in concentration in the post-surgery days were found. Again, with respect to the 1st day level: SCS: 0.89 ± 0.16 and HMP:1.02 ± 0.18, significant increases were found within each group as follows: SCS: at 3rd day 1.42 ± 0.42, 4th day 1.56 ± 0.44, 5th day 1.67 ± 0.41, and 6th day 1.54 ± 0.42; HMP: at 3rd day 1.51 ± 0.06, 4th day 1.53 ± 0.29, and 5th day 1.42 ± 0.30.

Finally, the comparison between the areas calculated under the curves (AUC) of the two groups resulted in being lower (ROS = −19%, TBARS = −28%, and PC = −7.6%) for the HMP than the SCS group. 

Finally, a positive correlation between the mean values of the creatinine levels and all the determined oxidative stress biomarkers in plasma could be established as follows: ROS (r = 0.73, *p* < 0.03), TBARS (r = 0.81, *p* = 0.001), and PC (r = 0.77, *p* = 0.03), see Figure 5.

### 2.4. ^1^HNMR and EPR Tissue Analysis

ROS concentration (μmol·g^−1^) calculated by EPR spectra collected from bioptic kidney tissue samples (0.0031 ± 0.0016 g, *n* = 14) at different experimental times—T0 (anesthesia), T1(end of warm ischemia -WI-, 75 min), and T2 (end of preservation, 8 h)—in SCS (full symbols) and HMP (empty symbols) are displayed in the histogram bars of Figure 6A.

Histogram bars showing the concentration (μmol/g) of the metabolites of greatest interest calculated from ^1^HNMR spectra collected on kidney tissue samples in SCS (full bars) and HMP (empty bars) groups are shown as well: lactate (Figure 6B), alanine (Figure 6C), valine (Figure 6D), and TMAO (Figure 6E).

With respect to T0, a great increase in the ROS levels (range 340–162%) was found in both groups at the end of ischemia (T1), and a great increase of ROS production was found in the two groups: SCS +525% and HMP 219+% at the end of perfusion (T2) as well. The difference between the two investigated groups at T2 was significant (*p* < 0.01). 

NMR experiments were carried out on a smaller number of samples (*n* = 6) due to the greater amount of tissue needed for NMR with respect EPR acquisitions. Therefore, only two experimental times (T0 and T2) were analyzed. For all the analyzed metabolites, a great increase of the concentration could be observed at T2 vs. T0. However, as is shown in the figure, the increase was lower for the tissues belonging to the HMP group as follows: lactate: −59.76%, alanine: −43.17%; valine: −58.56%; and TMAO: −77.96%.

### 2.5. Histological Evaluation

No changes were observed in either experimental animal tissue sections groups under light microscopy. Representative images of tissue samples obtained from kidneys belonging to SCS and HMP groups at T1 and T2 are shown in Figure 7. As is shown in the figure, light microscopy did not show any significant differences between the groups of kidneys preserved by SCS or HMP. In detail, the following were observed in both groups: at T2 (end of preservation), well-preserved glomeruli, patent, and dilated tubular lumina, and at Tend (sacrifice), focal interstitial mononuclear infiltration.

## 3. Discussion

As always occurs in medical science, early diagnosis and timely intervention will also improve outcomes in organ transplantation [43]. The association between DGF and worse outcomes has led to increased efforts to better understand the mechanisms of ischemia–reperfusion injury, the major cause of dysfunction and/or non-function upon reperfusion in the recipient, and to develop interventions to reduce its occurrence and impact [45,46].

Build-up of ROS and lactate secondary to anaerobic cellular metabolism, occurring during donor organ procurement and static cold storage, is the mediator of the ischemia–reperfusion injury [47]. Cellular swelling and acidosis are the consequences of the rapid depletion of ATP under anaerobic metabolism. This leads to adenosine degradation, causing the accumulation of toxic substances, such as hypoxanthine and xanthine oxidase generating ROS [48].

Nevertheless, oxygen free radicals are generated both during preservation and during reperfusion [49], stimulating oxidative damage and promoting apoptosis and necrosis. The severity of tissue injury, caused by hypothermic preservation, influences the level of I/R injury and the subsequent functionality of the kidney. Decreasing ROS, as observed by HMP, might increase successful kidney transplantations, resulting in breaking the vicious cycle of cell damage, inflammation, and distal organ impairment [50,51].

Metabolic support may be another important factor in the observed benefit of HMP vs. SCS for preserving kidneys prior to transplantation. Experimental studies detail the metabolic activity of kidneys under various storage conditions [28,29,42].

In the present study, the combined oxidative and metabolic approach achieved by the simultaneous use of EPR and NMR techniques confirmed previous findings demonstrating the benefits of HMP vs. SCS on chronic kidney graft outcome. 

It is believed that HMP protects against cold ischemic injury by providing a better flush of the kidney, clearing of red blood cells, and prevention of the accumulation of waste products. Compared with static conditions, it was demonstrated to support higher levels of metabolism [52,53].

These figures appear to be confirmed by the results of this study, as shown in Figure 2, Figure 3 and Figure 4. In fact, as can be observed in Figure 2, starting from almost the same level, during the HMP preservation period, ROS concentration linearly increased in the first two hours, then followed a low exponential decrease. TAC, in turn, showed an expected specular behavior, even though it was not of the same amount: ROS increased by 60%, while TAC decreased by 23%. By contrast, in the SCS group, during the entire preservation period, both ROS and TAC remained at a slightly lower level, reaching a significant difference (*p* < 0.001) at the end of the period. In accordance with that reported above, these findings suggest to us that the flush realized by the HMP promotes, over time, a gradual decrease of the ROS production, suddenly increasing in the first hours while, at the same time, allowing for a restoration of the TAC towards the initial level. One of the major factors in the pathogenesis of complications is the imbalance between the ROS formation and clearance by the antioxidative system. This disparity can cause endothelial dysfunction and impairment of the regulatory functions of endothelium for vasculation [54].

In this respect, the advantage of HMP vs. SCS is to eliminate from the organ the ROS amount produced during preservation. Indeed, ROS produced as a by-product of cellular respiration are important in many cell-signalling cascades, but their increased production leads to peroxidation of membrane lipids, proteins, and DNA damage, up to cellular dysfunction and death [55,56].

The advantage of the HMP vs. SCS found even more confirmation when the metabolic kidney response was considered, as is shown in Figure 3, where the absolute quantitative concentration trend of the most important metabolites is displayed. Alanine and valine, the most important metabolites indicators of proximal tubules disfunctions; acetate, whose level mainly increases in case of cortical lesions; trimethylamine-N-oxide (TMAO), a common metabolite in animals and humans, oxidation product of trimethylamine, and derived from Choline, indicator of medullar lesions; and finally, lactate, the most important indicator of an anaerobic cellular metabolic status and global ischemia [57] can be observed in the figure. 

A great difference between the two preservation methods is evident: the HMP machine allowed for a great amount of the metabolites, produced by the kidney itself, to flow out, thus washing waste products from the organ. An exponential increase of all self-produced metabolites was found, even though with a different rate constant. For each of them, an almost constant level can be observed for the SCS group. 

At the same time, in the HMP group, total GSH followed an exponential decay, while it remained almost at the same level in SCS. 

The antioxidant capacity of GSH, can be consumed when using the HMP machine, thus promoting a redox status restoration [58].

All the achieved results lead us to conclude that, unlike SCS, HMP restores, at least partially, the fundamental processes of glomerular filtration, which eliminate toxic solutes, including oxidative stress. 

On the other hand, delayed graft function is the most common complication in the immediate post-transplantation period, mainly in deceased renal allografts, almost invariably in the non-heart beating transplant [59]. The functional results of this study showed that, in the SCS group, there was a severe level of renal dysfunction with lower levels of creatinine clearance and an accentuated unbalance of redox homeostasis with higher levels of oxidative damage (see Figure 4). 

The observed significantly increased levels of lipid peroxidation (TBARS, Figure 4C) and protein carbonyls (PC, Figure 4D) were associated with significantly increased ROS production (Figure 4B). During the two weeks of observation, the transplanted animals presented a peak of ROS production and of damage biomarkers concentration at Days 3–4, remaining almost at the plateau level until Days 5–6, and then it decreased from Day 7 onwards. The time of the peak did not differ in the two monitored groups; however, the peak levels were significantly lower (range *p* < 0.01–0.0001), and the restoring kinetics were more rapid in the HMP group.

Lipid peroxidation is a product of the free-radical-mediated oxidation of arachidonic acid, which is associated with membrane damage and is known to instigate proinflammatory mediators and stimulate the release of cytokines and chemokines, thus causing tissue injury [60,61,62,63] and endoplasmic alteration and deterioration of tissue integrity after reperfusion [64].

At the same time, higher levels of protein carbonyl were found in the static storage groups compared with the other storage conditions. Other studies supported these findings, with increased oxidative damage [65,66].

Even more encouraging results were obtained by the EPR and NMR analysis carried out on bioptic kidney samples. Needless to say, the most important issue was to maximally preserve the integrity of the kidney, and, as is well known, the main limitation of NMR technique, compared with others, as for example EPR, is the sample amount needed to obtain the spectra. Therefore, for each kidney, one tissue sample was excised to be analyzed by EPR at T1 and T2, while only three samples for each group at T2 were biopted for NMR analysis. 

Nonetheless, the histogram bars of Figure 6 show an ROS increase at the different experimental times, but at T2, the ROS level resulted in being significantly lower (*p* < 0.01) in the kidneys perfused by the HMP machine compared with SCS. The same was observed for all the NMR analyzed metabolites, which showed an increase at T2 with respect to T0, but the increase was lower in the tissues perfused by the HMP machine. Interestingly, the trend was the opposite when compared with the perfusion liquid NMR analysis, as shown in Figure 3. This figure confirmed that when ROS and metabolites, inevitably increasing during the preservation time, are washed out by the HMP machine flow, their concentration logically increases in the perfusion liquid, but at the same time, lowers in the tissue with respect to the cold storage maintenance.

Finally, it is relevant to note, in the present study, that despite the lower ROS and TBARS productions in HMP vs. SCS and all the other measured parameters that indicated a better performance of HMP machine, the observed light microscopy images did not suggest a difference between the tissue samples excised from the two groups nor relevant impairment of the renal or tubule cell function during the 14-day post re-implantation phase (see Figure 7). These findings underline the relevance of the measurements performed by advanced techniques, such as EPR and NMR, to monitor transplanted organs during the entire period and even beforehand to be able to determine whether the organ will be reimplanted or discharged. On the other hand, the transplanted kidney would have been monitored for a longer period. 

In addition, in order to further enhance the effectiveness of perfusates, some additives to the solution could be used. Many substances are reported in the literature to counteract the ischemic injury by improving both the metabolic response to anaerobiosis and the oxidative stress. Oxygen and/or energy substrates can be supplemented to the perfusion solution. Particularly, oxygen carriers have been applied experimentally in kidney preservation. At the same time, the addition of free radical scavengers, such as superoxide dismutase (SOD), to the preservation solution has been found to be beneficial in preventing the generation of oxygen free radicals in this highly oxygenated environment [15,67]. 

## 4. Materials and Methods

### 4.1. Animal Model and Surgical Procedure

Fourteen healthy 30-kg female domestic pigs were sedated by an intramuscular injection of Zolazepam/Tiletamine 6 mg/kg. A general anaesthesia was induced (Induction: Propofol 2 mg/kg and Atracurium 1 mg/kg. Maintenance: Propofol 5–10 mg/kg/h and Atracurium 1 mg/kg bolus every 30 min. Analgesia: Diflunixin Meglumine 100 mg) after the placement of a peripheral catheter in a vein of the ear. Later, the surgical placement of a jugular central venous line took place, and a xyphopubic laparotomy was performed. The animals were intubated and mechanically ventilated with a 60% O_2_–air mix. The left renal artery and vein were isolated and clamped to mimic the condition of a DCD donor [68]. The clamping was maintained for 75 min (warm ischemia, WI); this timing was chosen to reach the maximum tissue damage, adopting an ischemic period much longer than the maximum warm ischemia time (WIT) allowed by the clinical protocols (40 min) [69]. The organ was then removed and immediately cold-flushed with 500–600 mL of preservation solution (Belzer Machine Perfusion Solution (MPS) or IGL-1) [70] at the constant pressure of 75 mmHg. Kidneys were preserved for 8 h at 4 °C and subdivided in two randomized groups (see Figure 7) as follows: Group 1: SCS (*n* = 7) by static conservation in IGL-1; Group 2: HMP (*n* = 7) using the RM3 Waters Medical Systems pulsatile machine in MPS. Pressure, flow, and resistance parameters were monitored by the perfusion machine. The systolic pressure was settled by the operator, then gradually increased during the first hour and stabilized until the end of the perfusion. Specifically, the starting pressure was set at 20 mmHg, then 2 mmHg was added every 8 min, reaching the maximal level of 35 mmHg at the end of the first hour. The flow would have ideally been 0.5 mL·g^−1^ tissue; it is insufficient for values less than 60 mL·min^−1^. Resistances had to be set in the range of 0.10–0.40 mmHg·mL^−1^·min^−1^. Renal blood flow (RBF) and systolic pressure (MAP) were recorded continuously. Intra-renal resistance (IRR) could be calculated as MAP/RBF. These cut-off values were established on the basis of both the experiments on animal models and the clinical experience of international transplant centres [44]. 

After 8 h of storage/perfusion, a renal orthotopic auto transplantation was performed by an end-to-side arterial anastomosis between the renal artery and the aorta and an end-to-side venous anastomosis between the renal vein and the inferior vena cava. Finally, the ureterovesical anastomosis (Lich–Gregoire technique) was performed. The contralateral right nephrectomy was carried out before the closure of the abdominal wall. All animals received humane care and all studies were carried out in accordance with policies and guidelines of the Italian Ministry of Health for the use and care of laboratory animals. All procedures were carried out under the animal use protocols approved by the ethical committee (IRCCS Policlinico San Matteo, Pavia, 27100 Italy, “Comitato Etico per la Sperimentazione Animale” 18/03/2014, N. 4/2014).

### 4.2. Post-Surgical Evaluation Procedure

The transplanted animals were kept under observation for 14 days and then sacrificed by an anaesthetic lethal injection procedure. During the first 7 post-operative days, analgesic (Ketorolac 15 mg), diuretic (Furosemide 10 mg), antibiotic (Cefotaxime 1 g), and gastroprotective (Omeprazole 20 mg) therapies were administered every 12 h. 

### 4.3. Sampling Methods

#### 4.3.1. Perfusate Sampling

For each kidney, 2 mL of perfusate was sampled, following the timing reported in Figure 7. The perfusate was transferred to a cryogenic vial and stored at −80 °C until NMR analysis. Samples were thawed only once for the analyses, which was performed within two weeks of collection. ROS production and total antioxidant capacity assessment in perfusate were performed immediately after collection. 

#### 4.3.2. Blood Sampling

Approximately 15 mL of blood was drawn from the jugular central venous line and collected in heparinized and serum vacutainer tubes (Becton Dickinson and Company, Oxford, UK). Plasma was separated by centrifuge (5702R, Eppendorf, Hamburg, Germany) at 3000× *g* for 5 min at 4 °C and immediately stored in multiple aliquots at −80 °C until assayed. Samples were thawed only once for the analyses, which was performed within two weeks of collection. SCr concentration assessment was performed immediately after collection.

#### 4.3.3. Tissue Sampling

The renal tissue was collected for histological, EPR, and NMR experiments after each of the following: anaesthesia (T0), warm ischemia (75 min) and flushing (T1), at the end of preservation (8 h, T2), and at the sacrifice (14th day: Tend).

### 4.4. Measurements

#### 4.4.1. EPR Measurements

An X-band EPR instrument (E-Scan—Bruker BioSpin, GmbH, Billerica, MA, USA) was utilized for ROS measurements. The instrument allows us to handle very low concentrations of paramagnetic species in small (50 μL) samples. Among the spin probe molecules suitable for biological utilization, the CMH (1-hydroxy-3-methoxycarbonyl-2,2,5,5-tetramethylpyrrolidine) probe was adopted.

ROS production rate was determined in perfusate and plasma samples by means of a well-established EPR method [31,32]. 

Acquisition EPR parameters were microwave frequency: 9.652 GHz; modulation frequency: 86 kHz; modulation amplitude: 2.28 G; center field: 3456.8 G; sweep width: 60 G; microwave power: 21.90 mW; number of scans: 10; and receiver gain: 3.17 × 10^1^. Sample temperature was firstly stabilized and then maintained at 37 °C by a temperature and gas controller “Bio III” unit, interfaced to the spectrometer. From the EPR spectra, relative quantitative determination of ROS production was obtained then converted into absolute concentration by using the stable radical CP* (3-Carboxy-2,2,5,5-tetramethyl-1-pyrrolidinyloxy) as external reference [71,72,73].

A small amount of tissue (0.0031 ± 0.0016 g, *n* = 14) was excised from each kidney of both groups at T0, T1, and T2 for EPR experiments. For ROS detection in kidney tissue, methods were previously described [34,35,74]. Briefly, samples were biopted, weighed, and immediately incubated at 37 °C in Krebs-HEPES buffer (KHB) with 25 μM deferoxamine methane–sulfonate salt (DF) chelating agent and 5 μM sodium diethyldithio-carbamate trihydrate (DETC) at pH 7.4 with 1 mM of spin probe CMH. After 30 min, the isolated tissues were placed in the center of a 1 mL plastic syringe according to Dikalov et al. [74], snap-frozen, and stored at −80 °C. Then, the frozen block was removed by gentle pushing from the warmed-up syringe and analyzed in the quartz Dewar with liquid N_2_. Spectra were recorded at 77 K; the acquisition parameters were modulation amplitude: 5 G; centered field: 2.0023 g; sweep time: 10 s; field sweep: 60 G; microwave power: 1 mW; number of scans: 10; and receiver gain: 1 × 10^3^. Data were, in turn, converted into absolute concentration levels (micromoles per gram) by adopting CP• stable radical as external reference.

Spectra were recorded and analyzed by using the Win EPR software (2.11 version) standardly supplied by Bruker. An example of the recorded EPR signal from perfusate and tissue showing the triplet coming from the interaction of the ^14^N–OH group of CMH with the ROS oxygen unpaired electron (NOH + O•_2_ → NO• + H_2_O_2_) is displayed in Figure 8A.

#### 4.4.2. Antioxidant Capacity 

The amount of 10 μL of perfusate was used to assess the reducing capacity by means of a commercial EDEL potentiostat electrochemical analyser (Edel Therapeutics, Lausanne, Switzerland). The instrument was equipped with a redox sensor in a three-electrode arrangement able to respond to all water-soluble compounds present in biological fluids and oxidized within a defined potential range [72,75,76]. Data were expressed in nW.

#### 4.4.3. Enzymatic Assays

##### Thiobarbituric-Acid-Reactive Substances (TBARS)

TBARS determination method was utilized to detect lipid peroxidation in plasma samples. By means of the TBARS assay kit (Cayman Chemical, Ann Arbor, MI, USA), a rapid photometric detection of the thiobarbituric acid malondialdehyde (TBAMDA) adduct at 532 nm was performed according to the manufacturer’s instruction. A calibration curve was obtained from pure malondialdehyde-containing reactions. 

##### Protein Carbonyls (PC)

The accumulation of oxidized proteins in plasma samples was measured by their reactive carbonyls content. A Protein Carbonyl assay kit (Cayman Chemical, Ann Arbor, MI, USA) was used to evaluate colorimetrically oxidized proteins as described in detail in the kit’s user manual. All tests were performed in duplicate, and all data were obtained by a microplate reader spectrophotometer (Infinite M200, Tecan, 8708 Männedorf, Switzerland).

#### 4.4.4. ^1^H-NMR Analysis

All ^1^H-NMR experiments were carried out at 400 MHz using a Bruker Avance (1H/BB BBI) vertical bore instrument with a z-field gradient accessory capable of delivering gradients up to 500 mT/m.

A selective Watergate sequence was used with sine-shaped gradients of 1 ms duration. The composite pulse was formed by a 3–9–19 binomial selective pulse (interpulse delay = 70 us). The p/2 hard and soft pulses were calibrated against the water signal. All perfusate solution (500 μL) samples collected were placed into a 5 mm glass NMR tube. Acquisition parameters were size: 32 K; spectral width: 20 ppm; acquisition time: 2.04 s; and relaxation delay: 1 s. Chemical shifts are reported from Sodium Trimethylsilyl Propionate (TSP, 0.2 mM) added to the solution as internal standard. All experiments were carried out at the controlled temperature of 28 °C. 

All NMR data are referred to the TSP signal, added in a known amount (10 mM) in 100% D_2_O solution into a capillary coaxially placed into the NMR tube for the acquisitions performed on the perfusion solution or added to the solution for tissue samples. Moreover, to monitor the kinetic trend of the metabolites during the 8 h of storage, the spectra were acquired every hour (HMP group) or every four hours (SCS group). The kinetical data could be obtained by scaling each spectrum to the TSP area. A specific NMR spectrometer automated calibration routine was used for this aim. This procedure was not applied to the metabolite signals acquired from tissues, where the data of each spectrum were calibrated against the TSP signal area.

Tissue samples (0.15 ± 0.03 g; *n* = 8) were excised from kidneys at T0 and T2, frozen, then thawed and gently placed into a 5 mm NMR tube. Phosphate Ringer solution in 100% D2O and TSP (10 mM) was added to completely fill the sensitive region of the coils. The same sequence and acquisition parameters (NS = 2048) were used. 

Resonances assignation was based on a combined extraction of polar and lipophilic metabolites obtained from cortical and medullar kidney samples (512 ± 36.77 mg) according to the Protocol by Beckonert et al. [77]: metabolic profiling, metabolomic, and metabonomic procedures for NMR spectroscopy of urine, plasma, serum, and tissue extract [77].

The extract samples were resuspended in 500 μL Phosphate buffer Ringer (100% D_2_O, TSP 1 mM), placed into a 5 mm NMR tube, and acquired by using the parameters used for the perfusion solutions (NS = 128). 

All the examined metabolite signals were quantified, at each perfusion time, by absolute integration of the NMR signals reported to the reference signal [38].

All spectra were processed with the standard TopSpin 2.1 Bruker software. All the experimental data were fitted with the Maquardt–Levenberg algorithm implemented in SigmaPlot 9.0 software (Systat Software Inc., San Jose, CA, USA).

Typical high-field ^1^HNNMR spectra collected on perfusate at the end of the perfusion time by MPH machine and from kidney tissue at the same experimental time are displayed in Figure 9B(a,b). The most relevant metabolites are indicated by arrows.

### 4.5. Histological Evaluation

At the T2 and Tend experimental times, corticomedullary kidney tissue was excised and fixed in 10% formaldehyde, dehydrated in an ascending grade of ethanol, cleared in xylene, and embedded in paraffin according to standard methods. The paraffin-embedded samples were cut into 3 μm sections for histopathological examination. Tissue sections were de-paraffinized in xylene and stained with hematoxylin and eosin (HE) (Sigma-Aldrich S.r.l., Milan, Italy, hematoxylin, MHS1; eosin, 230251), periodic acid-Schiff (PAS) (Sigma-Aldrich S.r.l., Milan, Italy, 395B), and methenamine silver–periodic acid-Schiff stain (Silver) (Sigma-Aldrich S.r.l., Milan, Italy PROTSIL1 Silver Stain Kit). Histopathological examinations of the kidneys were performed by an independent pathologist and one of the investigators. 

All tissue sections were examined by light microscopy (Nikon Eclipse E200). Glomerular sclerosis, mesangial and capillary wall changes, tubular, and interstitial and vascular pathologic lesions were investigated.

### 4.6. Renal Function Recovery Evaluation

To monitor the function recovery, blood samples were collected from the pig ear vein before the left kidney removal every day during the first week and at the 14th day after auto-transplantation. Serum was used to determine the creatinine levels using an Advia Chemistry XPT analyzer (Siemens Healthcare GmbH, Erlangen, Germany).

### 4.7. Statistical Analysis

Statistical analysis was performed using SPSS Statistics 17.0 (Software Inc., Chicago, IL, USA) and the GraphPad Prism version 9.3.1 for Mac OS X (Software Inc., San Diego, CA, USA). The Shapiro–Wilks W test was used to test variables for normal distribution.

Data were compared by one-way ANOVA for repeated measures, followed by Bonferroni’s multiple comparison test to further check the among-group significance.

The two-way analysis of variance (ANOVA) was run with “condition” (static cold storage: SCS; hypothermic machine perfusion: HMP) and “time” (for example, intra-operative from 0 to 8 h or post-surgery from 1st to 14th days). The Sidak’s multiple comparisons test was used to test the significance of the differences. Areas under the curve (AUC) by creatinine measurements were calculated for the entire monitoring period. Pearson product moment correlation coefficient (r) with 90% confidence intervals (CI) was used to examine the relationships among selected parameters (creatinine levels vs. oxidative stress biomarkers and/or the metabolite concentrations assessed by NMR versus the ROS production by EPR).

Quantitative data are presented as mean ± SD. A *p*-value < 0.05 was assumed as statistically significant. Change Δ% estimation (((postvalue − prevalue)/prevalue) × 100) is also reported in the text.

## 5. Conclusions

All these findings lead us to conclude that since the surgical procedure of transplantation and ischemic injury to the organ during the procurement and transplantation procedures cause increased oxidative stress, it seems that successful kidney transplant might result in better and more rapid normalization of the antioxidant status and lipid metabolism by eliminating free radicals and decreasing oxidative stress. These latter, joined with the possibility of flowing out metabolites and discharge products de novo produced by the kidney, might be valid reasons suggesting the perfusion by pulsatile machine, that has been shown to reduce the incidence of delayed graft function after transplantation of deceased donor kidneys from brainstem death (DBD) and circulatory death (DCD) donors [68,78], as the best preservation mode.

The promising results of the present study reached by the joint use of EPR and NMR techniques and enzymatic assays determinations encourage us to continue in this research direction.

## Figures and Tables

**Figure 1 ijms-24-01029-f001:**
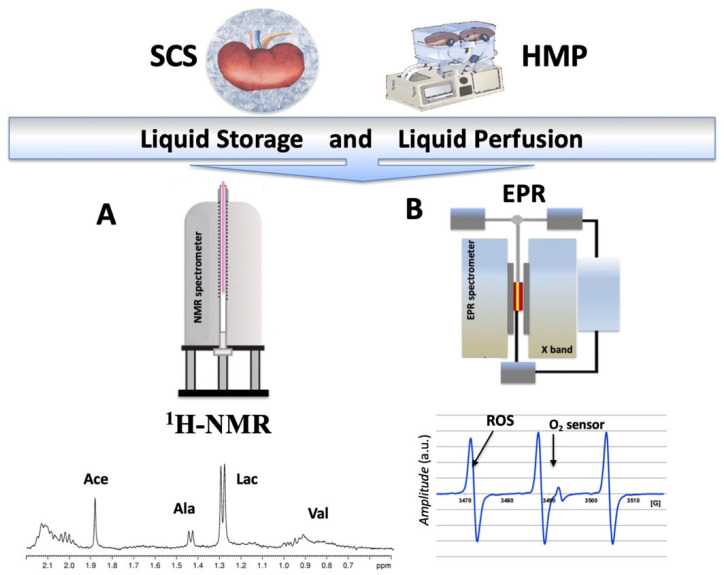
Static cold storage (SCS, the kidney stored in solution surrounded by crushed ice to keep the temperature between 4 and 0 °C is shown in the photo) and hypothermic machine perfusion (HMP, the kidneys in the machine are shown in the photo) preservation conditions assessed by NMR (**A**) and EPR (**B**) instruments. Under the instruments: (left) a typical high field ^1^HNMR spectrum obtained from the perfusion solution during a SCS or HMP experiment. The main assigned resonances are indicated by arrows: Valine: proximal tubules disfunction; Lactate: global ischemia; Acetate: cortical lesion; and TMAO: medullar lesion. Right: a typical EPR spectrum recorded from the perfusion solution at 37 °C. The triplet signal comes from the reaction of 1-hydroxy-3-carboxymethyl-2,2,5,5-tetramethyl-pyrrolidine spin probe (CMH, EPR silent) to 3-carboxymethyl-2,2,5,5-tetramethyl-pyrrolidinyloxy radical (CM, EPR active). When the signal is sequentially acquired, the ROS production rate can be calculated. Using a stable radical compound, such as 3-Carboxy-2,2,5,5-tetramethyl-1-pyrrolidinyloxy (CP), as a reference, the absolute concentration levels are obtained. The singlet is the signal from oxygen label (O_2_ sensor).

**Figure 2 ijms-24-01029-f002:**
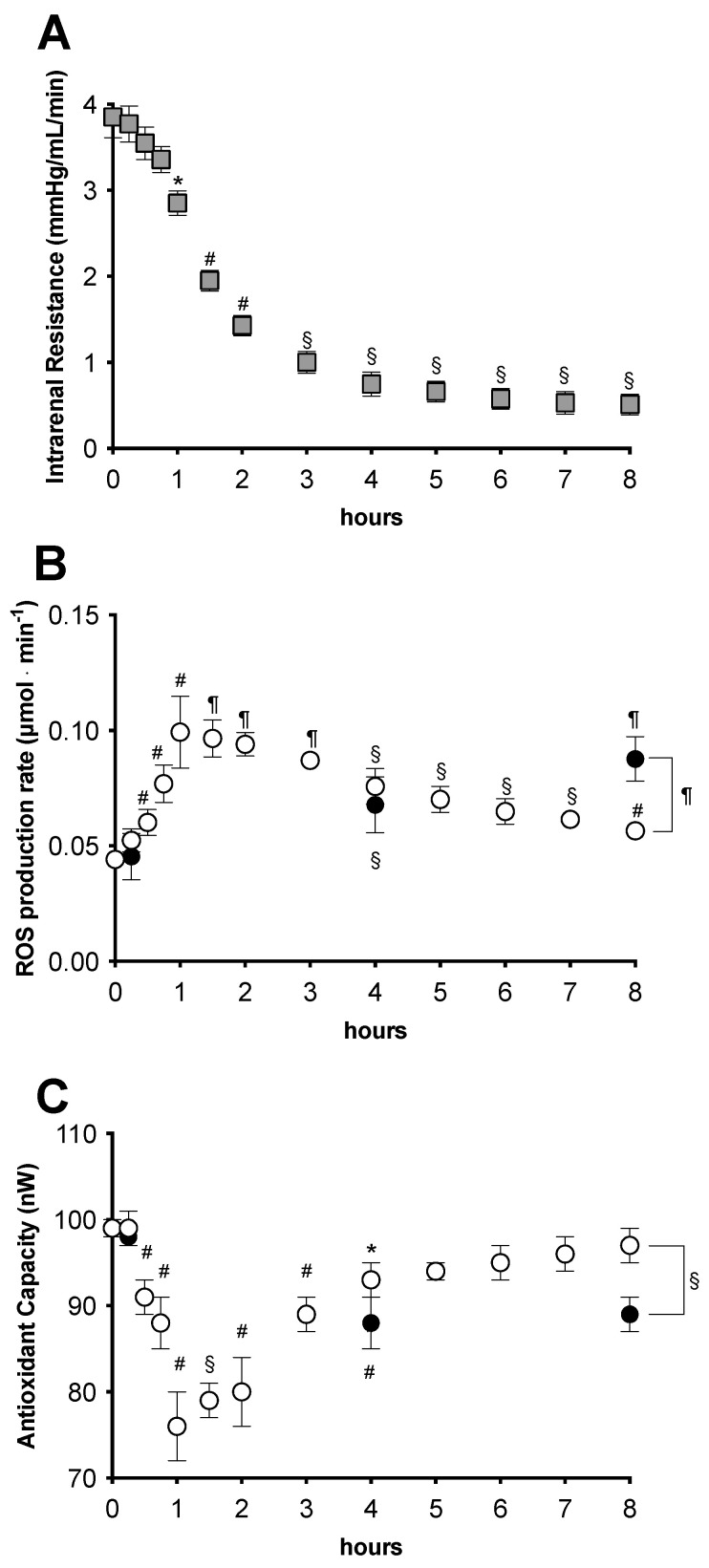
Intra-operative monitoring data. (**A**) Intra-renal resistance during hypothermic kidney machine perfusion (HMP, mmHg/mL·min); (**B**) EPR-measured reactive oxygen species production rate (ROS, μmol·min^−1^); and (**C**) Antioxidant capacity (TAC, nW) during static cold storage (SCS, full symbols) and hypothermic machine perfusion (HMP, empty symbols). The results are expressed as mean ± SD. Significant differences: * *p* < 0.05; # *p* < 0.01, § *p* < 0.001, and ¶ *p* < 0.0001.

**Figure 3 ijms-24-01029-f003:**
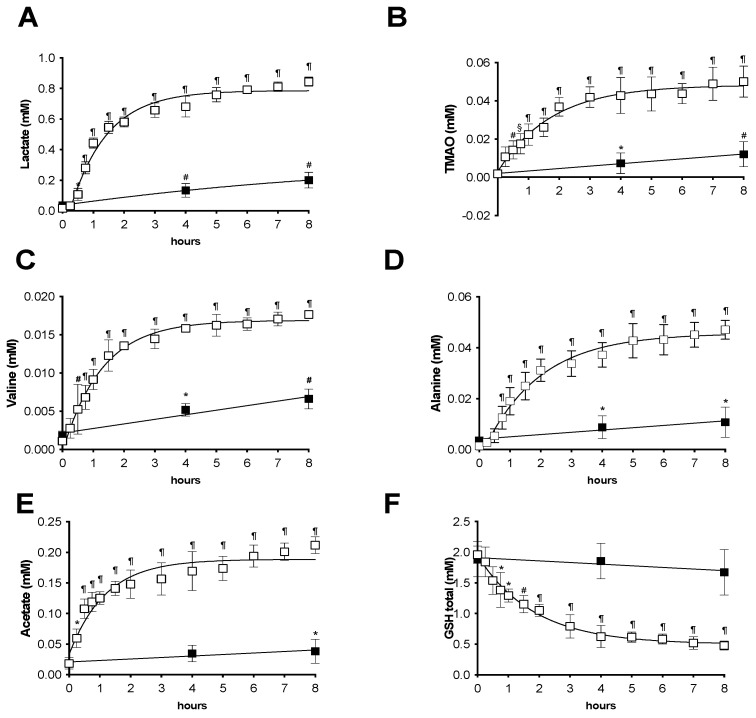
^1^H-NMR metabolite concentration (mM) during hypothermic kidney perfusion (HMP, empty symbols) and static cold storage (SCS, full symbols). The rate (mM/h) of the fitted trend resulted as follows: (**A**) lactate (HMP: K = 0.83, R^2^ = 0.98; SCS: K = 0.02, R^2^ = 0.97), (**B**) trimethylamine N-oxide (TMAO; HMP: K = 0.57, R^2^ = 0.98; SCS: K = 0.001, R^2^ = 0.99), (**C**) valine (HMP: K = 0.78, R^2^ = 0.99; SCS: K = 0.0005, R^2^ = 0.95), (**D**) alanine (HMP: K = 0.55, R^2^ = 0.98; SCS: K = 0.001, R^2^ = 0.90), (**E**) acetate (HMP: K = 0.85, R^2^ = 0.93; SCS: K = 0.002, R^2^ = 0.88), and (**F**) total glutathione (GSH; HMP: K = 0.55, R^2^ = 0.99; SCS: K = −0.026, R^2^ = 0.85). The results are expressed as mean ± SD. Significant differences: * *p* < 0.05; # *p* < 0.01, § *p* < 0.001, and ¶ *p* < 0.0001.

**Figure 4 ijms-24-01029-f004:**
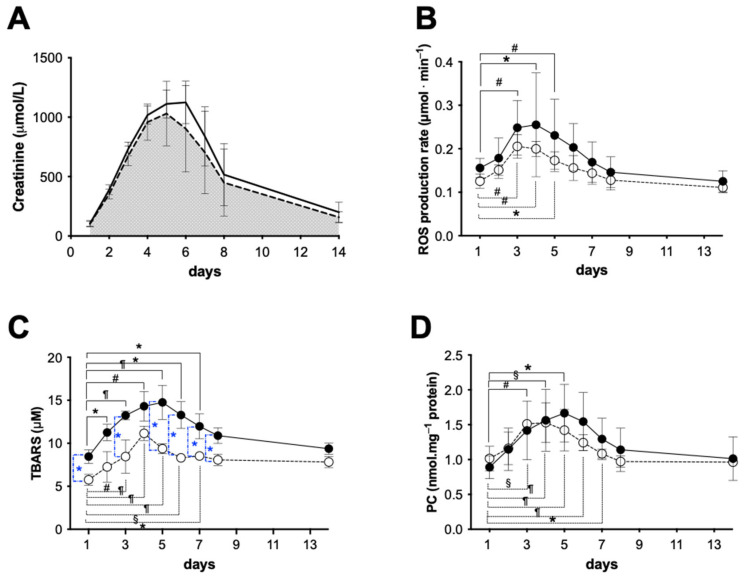
Renal functionality and oxidative stress after kidney re-implantation: (**A**) Serum creatinine concentration (mg/dL) measured from the HMP (dashed line) and SCS (continuous line) groups and areas under the curves (AUC); (**B**) Reactive oxygen species production rate (ROS, μmol·min^−1^) detected by EPR technique; (**C**) Thiobarbituric-acid-reactive substances (TBARS, μM); and (**D**) Protein carbonyl (PC, nmol·mg^−1^ protein) from 1st day to 14th day after kidney transplantation in static cold storage (SCS, full symbols) and hypothermic machine perfusion (HMP, empty symbols) groups. Continuous brackets indicate the significance of the intra-group data with respect to the first day. Blue stars (**C**) indicate the significance between HMP and SCS data at the same day. The data are expressed as mean ± SD. Significant differences: * *p* < 0.05; # *p* < 0.01, § *p* < 0.001, and ¶ *p* < 0.0001.

**Figure 5 ijms-24-01029-f005:**
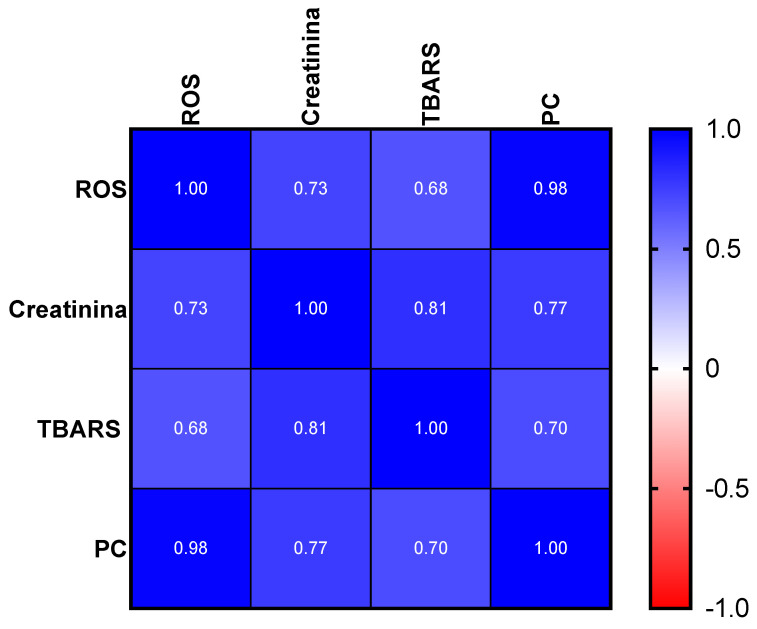
Heat map chart of creatinine levels and oxidative stress biomarkers in plasma: ROS, TBARS, and PC. The correlation coefficient (r) is reported in the squares.

**Figure 6 ijms-24-01029-f006:**
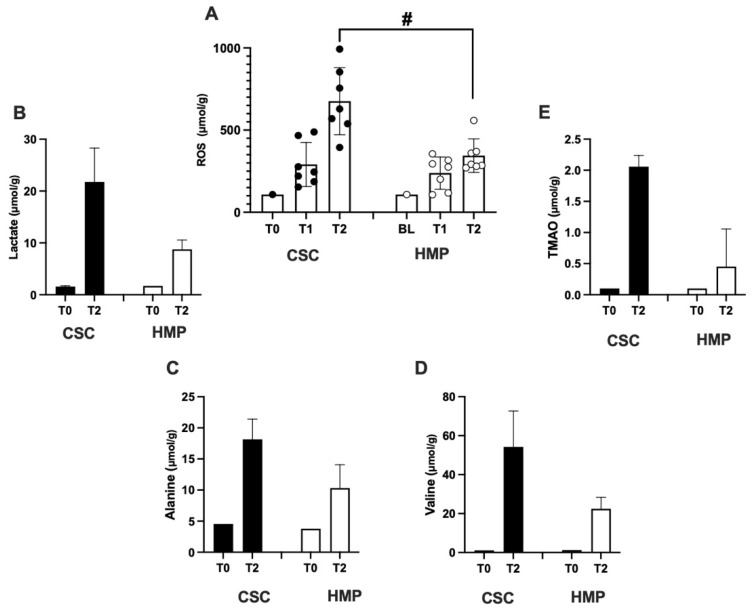
Histogram bars of: (**A**) ROS concentration (μmol/g) in kidney tissue at different experimental times: T0 (anesthesia), T1 (end of WI, 75min), and T2 (end of perfusion, 8 h) in SCS (full symbols) and HMP (empty symbols) groups. (**B**–**E**) metabolite concentration (μmol/g) calculated from ^1^HNMR spectra collected on kidney tissue samples in SCS (full bars) and HMP (empty bars) groups at T0 and T2 experimental times. Significant difference # *p* < 0.01.

**Figure 7 ijms-24-01029-f007:**
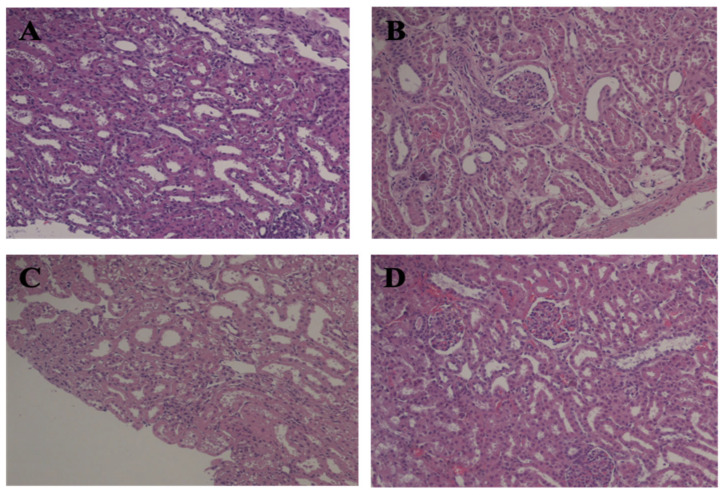
Histological examination. Representative light microscopy images of kidney tissue samples from SCS at T2 (**A**) and Tend (**B**) and HMP groups at T2 (**C**) and Tend (**D**). Magnification ×100.

**Figure 8 ijms-24-01029-f008:**
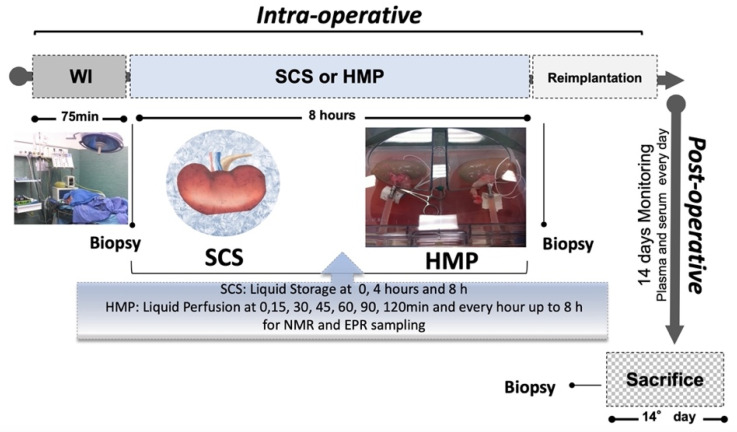
Sketch of the experimental protocol adopted to measure ROS production rate by EPR, metabolite concentration changes by ^1^H-NMR, and oxidative damage during the intra-operative (organ preservation) and post-surgery sessions. The storage solution was sampled at 0 (T1), 4, and 8 h (T2) in SCS and at 0, 15, 30, 45, 60, 90, 120 min, and every hour up to 8 h in HMP. Blood samples were collected every day until the 7th day after re-implantation and at the sacrifice (14th day: Tend). Kidney tissues were biopted after each of the following: anesthesia (T0), 75 min of WI (T1), after 8 h of preservation (T2), and at the 14th day (Tend: sacrifice). WI: warm ischemia; SCS: static cold storage; and HMP: hypothermic machine perfusion.

**Figure 9 ijms-24-01029-f009:**
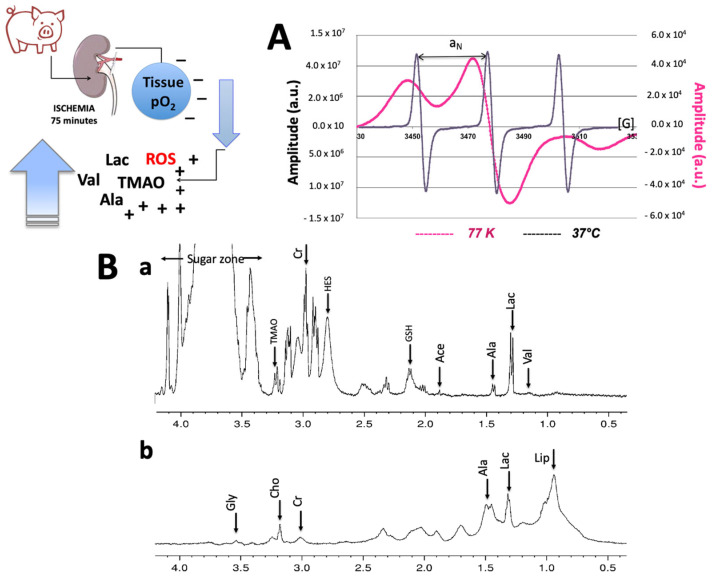
(**A**) EPR spectra recorded from both frozen kidney tissue (77 K) and from perfusion solution at 37 °C. The signals come from the reaction of 1-hydroxy-3-carboxymethyl-2,2,5,5-tetramethyl-pyrrolidine spin probe (CMH, EPR silent) to 3-carboxymethyl-2,2,5,5-tetramethyl-pyrrolidinyloxy radical (CM^_^, EPR active). (**B**) High-field ^1^HNMR spectra of (**a**) perfusate from a machine perfused in Belzer group at the end of the perfusion time (8 h, T2); (**b**) same expansion from a kidney biopsy at the same time. The correspondent metabolites are indicated by arrows.

## Data Availability

Data are available at request from the authors.

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
