# Peer review of "Evaluation of Oxidative Stress and Metabolic Profile in a Preclinical Kidney Transplantation Model According to Different Preservation Modalities"

_ijms, 2023, doi:10.3390/ijms24021029_

Round 1
Reviewer 1 Report
This is an interesting study aimed at evaluating by EPR and NMR spectroscopy the oxidative stress and the metabolic profile in ischemia/reperfusion injury, using two different preservation methods: cold static storage and ex vivo machine perfusion. The findings in this work suggest the ex vivo machine perfusion as the best method to preserve the organ compared to static preservation, resulting in a reduced ROS production.
However, I have some concerns regarding the NMR metabolomic analysis, that I report below.
1) Both for the perfusate and tissue, I see that the authors did not use any kind of data normalization. This could affect the changes of metabolites between the classes considered. I suggest the authors to clarify this point. For the preservation solution, can you normalize in respect to total integral?
Regarding tissue samples, can you normalize in respect to starting weight, as you did for the EPR analysis?
2) Did you collect other biofluids, such as urine or feces? These matrices can add more information regarding the metabolic status of the transplanted animal (e.g. collected before and after the transplant). In addition, both in urine and in feces there are some metabolites related to oxidative stress, such as hypoxanthine, guanosine-x- phosphate, adenosine-x-phosphate.
3) The authors collected serum samples; did you perform the serum NMR metabolomic analysis as well?
4) Did you correlate the NMR metabolite concentration to the EPR ROS concentration by Pearson or Spearman correlations? If so, this information could also be added.
5) I suggest the authors to perform a multivariate statistical analysis on NMR and EPR fused data, in order to verify the covariations between metabolic profiles and ROS.
Author Response
Comments and Suggestions for Authors
This is an interesting study aimed at evaluating by EPR and NMR spectroscopy the oxidative stress and the metabolic profile in ischemia/reperfusion injury, using two different preservation methods: cold static storage and ex vivo machine perfusion. The findings in this work suggest the ex vivo machine perfusion as the best method to preserve the organ compared to static preservation, resulting in a reduced ROS production.
The authors thank the reviewer for appreciating their study and for his helpful criticism. The point- by-point replies to the questions are reported below and the text changes highlighted in yellow in the revised version of the manuscript.
However, I have some concerns regarding the NMR metabolomic analysis, that I report below.
Both for the perfusate and tissue, I see that the authors did not use any kind of data normalization. This could affect the changes of metabolites between the classes considered. I suggest the authors to clarify this point. For the preservation solution, can you normalize in respect to total integral?
Regarding tissue samples, can you normalize in respect to starting weight, as you did for the EPR analysis?
R: The authors completely agree with the reviewer and thank him for the remark. NMR well as EPR techniques offer us the possibility of performing an absolute quantitative determination of metabolites and ROS, respectively. To this aim, a reference compound must be used: 3-Carboxy-2,2,5,5- tetramethyl-1-pyrrolidinyloxy (CP) for EPR and Sodium Trimethylsilyl Propionate (TSP) for NMR are the most suitable compounds, so that both EPR and NMR data are obtained following the same basic principles. Particularly, in the present study, all NMR data are referred to the TSP signal, added in a known amount (10mM) in 100% D2O solution into a capillary coaxially put into the NMR tube for the acquisitions performed on the perfusion solution samples or put externally for the experiments on the tissue. Moreover, to monitor the kinetic trend of the metabolites during the 8 hours of storage, spectra were acquired every hour (HMP group), every four hours (SCS group). The kinetical data could be obtained by scaling each spectrum to the TSP area. A specific NMR spectrometer automated calibration routine was used to this aim. This procedure was not applied to the metabolite signals acquired from tissues, where the data of each spectrum were calibrated against the TSP signal area. To clarify this point, the procedure was reported in the Material and Method section of the revised manuscript.
2) Did you collect other biofluids, such as urine or feces? These matrices can add more information regarding the metabolic status of the transplanted animal (e.g. collected before and after the transplant). In addition, both in urine and in feces there are some metabolites related to oxidative stress, such as hypoxanthine, guanosine-x- phosphate, adenosine-x-phosphate.
R: The authors agree with the reviewer. Unfortunately, the collection of other biofluids was not made for logistic reasons linked to the stabling of pigs and because it was not included in this study protocol. The authors anyway thank the reviewer for the good suggestion that can be considered in future researches.
3) The authors collected serum samples; did you perform the serum NMR metabolomic analysis as well?
R: Serum and plasma samples were collected for EPR, creatinine and oxidative stress damage biomarker assessments. The authors well agree with the reviewer that an NMR metabolomic analysis would be of great interest. Again, it was not included in the present study protocol, but could surely become matter of further investigation.
4) Did you correlate the NMR metabolite concentration to the EPR ROS concentration by Pearson or Spearman correlations? If so, this information could also be added.
R: The authors thank the reviewer for raising this interesting point. Following his suggestion, the positive significant correlations were found between EPR and each NMR metabolite concentration are now reported in the revised manuscript (see 2.2.1 section). Moreover, the Heat Map Chart of creatinine levels and oxidative stress biomarkers in plasma is now shown in the new Figure 5 (see 2.3.2. Oxidative stress evaluation section) and the procedure reported in 4.7 Statistical analysis section.
5) I suggest the authors to perform a multivariate statistical analysis on NMR and EPR fused data, in order to verify the covariations between metabolic profiles and ROS.
The authors thank this reviewer for the suggestion that might lead to interesting considerations. However as reported by Budaev, 2010, “….One important conclusion is that N=20 seems to represent the minimum sample size for the application of these multivariate methods” (Budaev, S. Using principal components and factor analysis in animal behaviour research: Caveats and guidelines. Ethology, 116, 472-480, 2010). More recently, other authors (Jenkins DG, Quintana-Ascencio PF (2020) A solution to minimum sample size for regressions. PLoS ONE 15(2): e0229345), reported that “…a minimum N = 8 is informative giving very little variance, but a minimum N ≥ 25 is required for more variance”. Based on this, we have not performed the multivariate statistical analysis on NMR and EPR data. On the contrary, significative correlations were calculated between EPR (ROS) data and each NMR metabolite. The results are reported in the revised version.
Reviewer 2 Report
The work presented by Mrakic-Sposta et al is interesting because it confirmed that the oxidative stress accompanied the ischemic injury of kidney graft during the transplantation process, from harvesting to reperfusion. This study also confirmed that the use of pulsatile machine allowing to perfuse and therefore wash kidney graft of oxidative metabolites, reduced the incidence reperfusion injury after transplantation.
The conclusions of the present paper are not really new, apart from the fact that the authors took care to accurately measure the production of ROS during the process of renal IR, using an NMR and EPR approach.
My questions and comments are as follows:
1- The abstract needs to be rebuilt and rebalanced. Indeed, the first sentence for example is, in my opinion irrelevant. The acronyms NMR and EPR appear for the first time without explanation. The results and the conclusion are contained in a single sentence (last 2 lines) : it’s too short.
2- Conversely, the introduction seems long to me (77 lines). Although very interesting to read, some paragraphs are off topic with respect to the objective of this paper. I think it would be useful to refocus the text.
3- The authors insist (rightly) on the need to develop “methodologic criteria leading to the decision about the reimplant suitability or discharging” (line 59). My question is therefore: is it possible in a clinical situation to consider the use of NMR and EPR methods to assess the state of oxidation of the graft before transplantation, or is this a experimental research tool only?
Based on their data, can the authors establish a correlation between the level of oxidative stress before transplantation and the outcome of the kidney graft after transplantation? Can the authors establish that from a certain level of oxidative stress the preserved graft is at risk and becomes non-transplantable? Have the authors taken any observations in this direction?
4- Figure 1. Some legends are not visible. We do not understand what the photo of SCS represents. The text must be completed and rewritten: the analyzes on SCS are not mentioned. The acronym CP is not explained.
5- Figure 2. What are the “open symbols” and “closed symbols”. Do you mean white or empty circle for open symbol?
6- Figure 3. How the authors explain that the release of metabolites (lactate, TMAO, Acetate, etc.) is about 10 times lower in the perfusion medium than in the kidney tissue (figure 5). If we follow the logic of the authors, the metabolites produced by the renal tissue medium should gradually increase during perfusion. Why does the production of metabolites reach a plateau after 3 hours of perfusion?
7- To demonstrate that oxidative stress is the essential cause of renal dysfunction, it would have been useful to provide groups (SCS and HMP) with or without antioxidants and to show that the presence of antioxidants blocks the observed deleterious effects. Do the authors have any comments/observations in this regard?
Author Response
Comments and Suggestions for Authors
The work presented by Mrakic-Sposta et al is interesting because it confirmed that the oxidative stress accompanied the ischemic injury of kidney graft during the transplantation process, from harvesting to reperfusion. This study also confirmed that the use of pulsatile machine allowing to perfuse and therefore wash kidney graft of oxidative metabolites, reduced the incidence reperfusion injury after transplantation.
The conclusions of the present paper are not really new, apart from the fact that the authors took care to accurately measure the production of ROS during the process of renal IR, using an NMR and EPR approach.
The authors thank the reviewer for appreciating their study and for his helpful criticism. The point- by-point replies to the questions are reported below and the text changes highlighted in grey in the revised version of the manuscript.
My questions and comments are as follows:
1-The abstract needs to be rebuilt and rebalanced. Indeed, the first sentence for example is, in my opinion irrelevant. The acronyms NMR and EPR appear for the first time without explanation. The results and the conclusion are contained in a single sentence (last 2 lines): it’s too short.
R: Following this reviewer’s suggestions, the abstract was rebuilt and rebalanced: results and conclusion are now better presented, while unnecessary items eliminated. The acronyms NMR and EPR were explained.
2- Conversely, the introduction seems long to me (77 lines). Although very interesting to read, some paragraphs are off topic with respect to the objective of this paper. I think it would be useful to refocus the text.
The authors thank the reviewer for appreciating and agree with him as regard as the excessive length of the Introduction that was shortened in the revised version.
3- The authors insist (rightly) on the need to develop “methodologic criteria leading to the decision about the reimplant suitability or discharging” (line 59). My question is therefore: is it possible in a clinical situation to consider the use of NMR and EPR methods to assess the state of oxidation of the graft before transplantation, or is this a experimental research tool only?
R: By the E-scan X-band EPR spectrometer, the instrument adopted in this study, a real-time quantification is possible. This instrument responds to the features of easy portability and handling. Moreover, it allows to deal with very low concentration amounts of paramagnetic species in small (50 ?L) samples, while the assessment procedure needs about 15 min (preparation, spectra acquisition and analysis).
As regard as the possibility of an NMR metabolomic analysis, the use of an ultra-high field NMR spectrometer to further increase sensitivity and specificity, at the same time reducing the acquisition time, with respect to the field (11.4 T) used in this study would be the gold standard. The procedure could be accompanied by a ultra-high field MRI morphological/functional kidney analysis
Based on their data, can the authors establish a correlation between the level of oxidative stress before transplantation and the outcome of the kidney graft after transplantation?
Can the authors establish that from a certain level of oxidative stress the preserved graft is at risk and becomes non-transplantable?
Have the authors taken any observations in this direction?
The sample size analyzed in the present study is too little to be able to establish certain levels and/or correlation between oxidative stress level and graft outcome. At the moment, we would hazard that the lower ROS and TBARS production observed during the survival period after HMP vs SCS preservation might be related to lower ROS production and higher levels of waste products found in preservation solution. At the same time, EPR levels significantly correlate to NMR metabolite concentrations. The numerous observations (about 1000) performed by our researching team on human plasma allowed us to reach a precise idea of the normal range of ROS production and oxidative stress, so that it would be possible to establish a ‘cut-off-line’. Thus, by increasing the number of the specific samples examined (pig kidneys) we can hypothesize that it would become likewise possible to reach the same result. So far, the results are promising and encourage us to continue in this researching direction.
4- Figure 1. Some legends are not visible. We do not understand what the photo of SCS represents. The text must be completed and rewritten: the analyzes on SCS are not mentioned. The acronym CP is not explained.
The authors agree with the reviewer and thank him for the observations. The Figure was redrawn: the authors are now confident that the reader will understand that the image shows a kidney. The text was rewritten, adding the missing information.
5- Figure 2. What are the “open symbols” and “closed symbols”. Do you mean white or empty circle for open symbol?
The authors agree. ‘White symbols’ instead of ‘open symbols’ is used in the revised manuscript.
6- Figure 3. How the authors explain that the release of metabolites (lactate, TMAO, Acetate, etc.) is about 10 times lower in the perfusion medium than in the kidney tissue (figure 5).
The authors thank the reviewer for his remark. Indeed, the data reported in Figure 6 are referred to 1 gram of tissue. On the contrary, the concentration data assessed in the preservation solutions (Figure 3) are referred to 1 liter of solution. In particular, the tissue metabolite amounts are not sampled from the renal vein but from the tank containing the perfusion solution (1L capacity). This is the main reason why they result so diluted.
If we follow the logic of the authors, the metabolites produced by the renal tissue medium should gradually increase during perfusion. Why does the production of metabolites reach a plateau after 3 hours of perfusion?
We hypothesize that 3 h is the time necessary to elute the metabolites produced and accumulated in the tissue during the warm ischemia. Thereafter, the accumulation is drastically reduced, due to the low tissue metabolism under the preservation conditions (low temperature). The ROS production rate follows almost a parallel trend.
7- To demonstrate that oxidative stress is the essential cause of renal dysfunction, it would have been useful to provide groups (SCS and HMP) with or without antioxidants and to show that the presence of antioxidants blocks the observed deleterious effects. Do the authors have any comments/observations in this regard?
R: Both preservation solutions utilized in the present study (commercially available products) contain antioxidants (i.e. Glutathione and Allopurinol). Nevertheless, the authors fully agree with the reviewer: in order to further enhance the effectiveness of perfusates some additives to the solution can be used. Many substances are reported in the literature to counteract the ischemic injury, by improving both the metabolic response to anaerobiosis and the oxidative stress. Oxygen and/or energy substrates can be supplemented to the perfusion solution. Particularly, Oxygen Carriers like Hemarina-M101 have been applied experimentally in kidney preservation. At the same time, the addition of free radical scavengers such as superoxide dismutase (SOD) to the preservation solution has been found to be beneficial in preventing the generation of oxygen free radicals in this highly oxygenated environment. The authors thank this reviewer for his relevant observation. Some considerations about can be found in the revised version.